# Cryo-EM structure of bixafen-bound *S. cerevisiae* complex II unravels SDHI specificity against pathogenic fungi
Nikos Pinotsis [1,2] ✉, Claudia Burn-Leefe[1], Sarah Jones[2], Shu Chen[2], Natalya Lukoyanova [2,5], Brigitte Meunier [3], Edward A. Berry [4] & Amandine Maréchal [1,2] ✉

Respiratory complex II (CII), or succinate dehydrogenase, couples succinate oxidation in the Krebs cycle with electron transfer to the respiratory chain. Owing to this pivotal role, CII inhibitors are widely used fungicides globally; however, their development has largely proceeded without structural insights from fungal targets. Here, we report cryo-electron microscopy structures of the 128 kDa mitochondrial CII from *Saccharomyces cerevisiae* in two states: active, with endogenous ubiquinone-6 bound (3.15 Å), and inhibited with the fungicide bixafen (3.00 Å). Although closely related to the mammalian type C enzyme, our structures show that the yeast CII has lost the canonical heme cofactor. They also reveal how clade-specific sequence extensions of the membrane subunits Sdh3 and Sdh4 - conserved in pathogenic fungi - uniquely contribute to complex stability and fungicide binding. Our findings provide a foundation for rational design of next-generation CII inhibitors and combatting resistance, in both agriculture and human health.

Cells need a permanent supply of energy to power the myriad of reactions that keep them alive. Most of this comes from cellular respiration and the ability of cells to convert food into adenosine triphosphate (ATP), the universal energy currency.

The main source of energy for cellular respiration, whether anaerobic or aerobic, is glucose. Glycolysis followed by pyruvate oxidation and the Krebs cycle, generate some of the ATP and the essential substrates to drive respiratory chains and oxidative phosphorylation. Ultimately, most cellular ATP is produced by the concerted action of the respiratory complexes embedded in the cytoplasmic membrane of bacteria or, in eukaryotes, the inner membrane of mitochondria (IMM)[1].

A wealth of information on the mechanism and membrane organization of the main respiratory complexes, complexes I-V, has been derived from structural biology and the use of model systems such as the cost-effective, uniquely genetically amenable, yeast *Saccharomyces cerevisiae*. The respiratory chain of *S. cerevisiae* is very similar to that of mammals except that it lacks complex I. Instead, it has non proton motive mitochondrial NADH dehydrogenases which are bound to both external (Nde1 and 2) and internal (Ndi1) sides of the IMM[2,3]. Today complete atomic models are available for all yeast respiratory complexes[4–6], as well as the supercomplexes they can form[4,7,8], except complex II.

Respiratory complex II (CII) is a central enzyme in cellular metabolism[9]. It catalyzes the reversible oxidation of succinate to fumarate as part of the Krebs cycle and funnels the electrons to the membrane quinone pool for oxidative phosphorylation, effectively linking the two major metabolic pathways. It is thus a succinate:quinone reductase (SQR), embedded within the cytoplasmic membrane of many bacteria and the inner membrane of all mitochondria, and the only membrane bound enzyme of the Krebs cycle. It is thought to have evolved from a soluble succinate dehydrogenase (SDH) that attached itself to the membrane to allow electron transfer from succinate to quinone[10]. Unlike respiratory complexes I, III and IV, CII does not translocate protons across the membrane but indirectly contributes to the formation and maintenance of the essential proton motive force by providing electrons to the quinone pool. More recently, there has been evidence for a wider implication of CII in hypoxia[11] and succinate signaling[12] which has raised interest on the true extent of its physiological role[9].

Structurally, SQRs and homologous quinol:fumarate reductases (QFRs) form a superfamily of enzymes that spans all three domains of life and display a common core of 3–4 protein subunits[13]. Two subunits (in mammals SDHA and SDHB) form the hydrophilic catalytic domain, that protrudes into the cytoplasm or mitochondrial matrix. The first hydrophilic

[1]Department of Structural and Molecular Biology, University College London, London, UK. [2]Institute of Structural and Molecular Biology, Birkbeck College, London, UK. [3]Université Paris-Saclay, CEA, CNRS, Institute for Integrative Biology of the Cell (I2BC), Gif-sur-Yvette, France. [4]Biochemistry and Molecular Biology, SUNY Upstate Medical University, Syracuse, NY, USA. [5]Present address: Structural Biology Science Technology Platform, Francis Crick Institute, London, UK. ✉e-mail: n.pinotsis@ucl.ac.uk; a.marechal@ucl.ac.uk

subunit (SDHA) is a flavoprotein where the substrate binds and exchanges two electrons with a flavin adenine dinucleotide (FAD) cofactor which in SQRs is covalently bound. The other hydrophilic subunit (SDHB), also called the iron-sulfur protein, contains three sequential iron-sulfur clusters that transfer the electrons one at a time between the FAD cofactor and the quinone. The other one or two subunits (in mammals SDHC and SDHD) form the membrane anchor and, together with residues from SDHB, the quinone binding site (Q site) which is located at the membrane interface with the cytoplasmic or matrix side.

Based on divergence in their membrane domain and associated cofactors, the members of the superfamily have been classified into six types A-F[10,14] with atomic models today available for representatives of all but type E[15–19].

To our understanding, all mitochondrial CIIs have been classified as belonging to type C which is characterized by a transmembrane domain made of at least two subunits that each provide an amino acid as axial ligand for the binding of one bis-ligated heme B (in mammals, SDHC and SDHD each provide a histidine). This is supported by all available mitochondrial structures solved to date[20–23] with the exception of recently characterized divergent CIIs within supercomplexes isolated from the ciliate *Tetrahymena thermophila*[24,25] and parasite *Perkinsus marinus*[26] which lack that heme cofactor in their membrane domain altogether.

Because of its central role in eukaryotic cellular metabolism, CII has been the object of intense research efforts in different fields such as medicine and agriculture. Mutations within human CII have been linked to a range of diseases including cancer[27,28] while the development of inhibitors of the enzyme has attracted considerable clinical and commercial interest with applications in human health[29–31] and food production[32,33]. Nevertheless, targeting mitochondrial enzymes raises questions about the assessment of mitotoxic risks[34] while mutations of CII in pathogenic fungi are emerging as a serious fungicide resistance threat in global agriculture[35].

Q site inhibition is the mechanism behind the important class of fungicides succinate dehydrogenase inhibitors (SDHIs). Having been in use since the 1960s, at least 20 species have acquired resistance mutations to SDHIs[36]. While most residues of the Q site are highly conserved, some are more variable, and could account for the specificity of Q site inhibitors[37]. It is noteworthy, however, that no fungal CII structure has been solved to date. The binding of several SDHIs at the Q site of CII from different forms of the mitochondrial enzyme has been investigated by X-ray crystallography[37,38] but the exact binding of many SDHIs including bixafen, a SDHI released in 2011 with widespread use in agriculture controlling pathogens of cereals, fruit trees, vegetables or field crops[32,33], is yet to be experimentally determined.

CII of the fungal yeast *S. cerevisiae* has been characterized genetically, and the sequences of the four subunits Sdh1-4 (homologs of mammalian SDHA-D) have been determined[39–42]. However to date, no successful isolation of the enzyme has been reported and outstanding questions, such as the presence of a heme in the complex, remain unanswered[43].

We report here the cryo-EM structures of the yeast CII in the presence of endogenous quinone and with the fungicide bixafen bound. Structures were resolved at 3.15 Å and 3.00 Å resolution, respectively, confirming the absence of heme B in the membrane domain of the yeast enzyme, though homology in structure and quinone reduction site confirms its classification into type C. The structures reveal a well-defined Q site and provide a detailed picture of the binding of bixafen. This work demonstrates the potential of the yeast *S. cerevisiae* to be developed as a cost-effective system to design and test the next generation of SDHIs.

## Results
### CII purification and activity measurements
To facilitate purification of CII, a yeast strain was engineered to harbor a ten-histidine tag at the C-terminus of Sdh4. The cells were cultured in ethanol medium, harvested in mid-log phase, and their mitochondrial membranes were isolated by mechanical means as described in Methods. The membranes were solubilized at a 10:1 ratio (w:w) of glyco-diosgenin (GDN) to

total protein content and CII was purified by successive steps of affinity chromatography and size exclusion chromatography (Supplementary Fig. 1a).

A reduced-minus-oxidized visible absorption spectrum recorded on the purified enzyme confirmed the presence of a FAD co-factor (Supplementary Fig. 1b). The FAD concentration in the final sample (5.4 µM) was similar to the total protein content determined by BCA assay (0.64 mg/mL or 5.0 µM), consistent with a pure preparation of CII where the flavin cofactor is stoichiometrically bound. No signature attributable to a heme group could be detected in the visible range. The purity of the CII preparation was also confirmed by BN-PAGE and CN-PAGE followed by in-gel activity assay (Supplementary Fig. 1c,d), which showed a single band migrating at an apparent molecular weight of 240 kDa when calibrated with soluble proteins. This is greater than CII's molecular weight of 128 kDa but not inconsistent with a monomeric form of the enzyme as the migration of small membrane proteins in native gels is strongly influenced by the lipid and detergent molecules they carry[44].

The purified enzyme was assayed with DCIP and displayed an electron transfer rate of $8.0 \pm 0.9$ e.s$^{-1}$ (Supplementary Fig. 1e, f). Addition of malonate (5 mM) or bixafen (5 µM), which target the succinate or quinone binding site, respectively, led to complete inhibition. The IC$_{50}$ of bixafen on the purified CII preparation was estimated to be slightly below 0.5 µM (Supplementary Fig. 1g). This is within the range of IC$_{50}$ values reported in the literature for bixafen on other fungi CII, from 0.03 to 0.1 µM for *Zymoseptoria tritici*[45] to 2.95 µM for *Rhizoctonia solani*[46]. Taken together, these results support the intactness and functional integrity of our yeast CII preparation.

### Structure of the *S. cerevisiae* complex II
The structure of the *S. cerevisiae* CII was determined by cryo-EM in two states: active, as purified, referred to as the native structure (CII-nat), and following incubation with bixafen (CII-bix) (Table 1, Fig. 1 and Supplementary Figs. 2 and 3). Resolved at 3.15 Å and 3.00 Å (Supplementary Figs. 4 and 5), respectively, the two structures superimpose with an overall root mean square deviation (RMSD) of 0.36 Å (1015/1040 matched residues) suggesting a virtually identical fold, with only minor differences around the Q site where bixafen binds (discussed below). Unless otherwise specified, the CII-bix structure is used for structural analysis given its highest overall resolution.

The yeast enzyme, like CII from metazoan and proteobacterial sources, consists of four subunits. Sdh1 and Sdh2, homologs of mammalian SDHA and SDHB, are hydrophilic and form the catalytic domain while Sdh3 and Sdh4, homologs of SDHC and SDHD, are hydrophobic and serve to anchor CII to the membrane.

The *S. cerevisiae* flavoprotein Sdh1 contains a covalently bound FAD cofactor (Supplementary Fig. 6) and is structurally conserved to the human (PDB ID 8GS8) and avian (PDB ID 6MYO) homologs with RMSDs of 0.96 Å (499/614 matched residues) and 1.06 Å (503/612 matched residues), respectively. It comprises four domains: the FAD-binding domain, the cap domain, a helical domain and a C-terminal domain (Fig. 2a).

The major structural difference of the yeast CII Sdh1 compared to the human and avian counterparts is found on the cap domain (residues 285–399). In our structures, no ligand targeting the succinate/fumarate binding site was added during sample preparation and the cap domain is found partially disordered. When the ligand binding site – at the interface of the FAD and cap domains – is empty, the cap domain residue Arg331 interacts with histidines 287 and 398, triggering conformational changes in the entire cap domain (Fig. 2a, left inset). As described in Supplementary text (Flavoprotein Sdh1), there is some evidence that the cap domain may actually be in a partially open position in our yeast structures.

In the proximity of the succinate/fumarate binding site, our EM maps show density for a metal which appears to have a stabilizing role (Supplementary Fig. 6). We modelled a potassium ion based on ligand geometry and bond lengths[47] (Supplementary text, Flavoprotein Sdh1) as well as analogous modelling in the structure of the avian homologue[21]. The side

chain of Tyr399 is involved in the potassium octahedral coordination while forming a π-stacking interaction with Arg444, which is part of the succinate/fumarate binding site. The main chain of Glu433 interacts both with the potassium ion and the FAD cofactor (Fig. 2a, right inset).

The Sdh2 iron-sulfur subunit (Fig. 2b) is made of two domains, an N-terminal domain resembling plant ferredoxins with one [2Fe-2S] cluster (residues 31-136), and a C-terminal domain resembling bacterial ferredoxins with one [4Fe-4S] and one [3Fe-4S] cluster (residues 137–266). This

## Table 1 | Cryo-EM data collection, refinement and validation statistics

| | CII-nat (EMDB-53029) (PDB 9QDL) | CII-bix (EMDB-53030) (PDB 9QDM) |
|---|---|---|
| **Data collection and processing** | | |
| Magnification | 105,000 | 105,000 |
| Voltage (kV) | 300 | 300 |
| Electron exposure (e⁻/Å²) | 49.5 | 49.5 |
| Defocus range (μm) | −2.7 to −1.2 | −2.7 to −1.2 |
| Pixel size (Å) | 0.828 | 0.828 |
| Symmetry imposed | P1 | P1 |
| Initial particle images (no.) | 5,949,939 | 5,066,550 |
| Final particle images (no.) | 149,808 | 146,416 |
| Map resolution (Å) at 0.143 FSC threshold | 3.15 | 3.00 |
| Map resolution range (Å) | 5.0–2.7 | 5.0–2.7 |
| **Refinement** | | |
| Initial model used (AF codes) | Q00711,P21801, P33421, P37298 | |
| Model resolution (Å) 0.143 FSC threshold | 3.03 | 2.95 |
| Model resolution range (Å) | 281.52–2.62 | 281.52–2.50 |
| Map sharpening *B* factor (Å²) | 60.38 | 64.88 |
| Map CC (masked) | 0.8037 | 0.8015 |
| Model composition | | |
| Non-hydrogen atoms | 8131 | 8258 |
| Protein residues | 1018 | 1040 |
| Waters | 18 | 27 |
| Ligandsᵃ | 8 | 8 |
| *B* factors (Å²) | | |
| Protein | 110.52 | 104.32 |
| Solvent | 84.34 | 100.26 |
| Ligand | 101.34 | 102.53 |
| R.m.s. deviations | | |
| Bond lengths (Å) | 0.005 | 0.004 |
| Bond angles (°) | 1.036 | 0.532 |
| Validation | | |
| MolProbity score | 1.75 | 1.81 |
| Clashscore | 7.63 | 9.88 |
| Poor rotamers (%) | 0.71 | 0.35 |
| Ramachandran plot | | |
| Favored (%) | 95.20 | 95.82 |
| Allowed (%) | 4.70 | 4.18 |
| Disallowed (%) | 0.10 | 0.00 |

ᵃFor CII-nat: FAD, K + , F3S, SF4, FES, 3PE, FAD, UQ6.
For CII-bix: FAD, K + , F3S, SF4, FES, 3PE, FAD, A1I6X (Bixafen).

is the most conserved subunit of the complex with an RMSD to the human and avian homologues of 0.68 Å (231/236 matched residues) and 0.76 Å (231/236 matched residues), respectively. All three iron-sulfur redox centers are well positioned in the cryo-EM maps (Supplementary Fig. 6) with separation distances, also to FAD and UQ6 (as modelled in the CII-nat structure) or bixafen (in CII-bix), similar to those seen in human, avian and bacterial enzymes, and consistent with rapid electron transfer during catalysis[15,17,21]. More structural details are given in the Supplementary text (Iron-sulfur protein Sdh2).

The membrane anchor of CII is evolutionarily more recent than the extrinsic hydrophilic subunits Sdh1 and Sdh2[10] and shows much greater diversity between homologues from different species. In *S. cerevisiae*, it is a heterodimer of two small subunits (Sdh3 and Sdh4) with similar fold, that may have arisen from a homodimer by a gene duplication event[13]. Each subunit has three transmembrane helices (TMHs, Fig. 3a, b and Supplementary Fig. 7). The first two TMHs from each subunit (α2, α4 from Sdh3 and α1', α2' from Sdh4) come together to form a four-helix bundle that in type C SQRs normally encloses one heme B, coordinated by two histidine residues. While the histidine at position 156 of TMH α4 of Sdh3 is conserved in *S. cerevisiae*, there is a tyrosine at the position 108 of TMH α2' in Sdh4, and no heme can be seen in our EM maps (see below) (Fig. 3a, b and Supplementary Fig. 8), consistent with spectroscopic analyses (Supplementary Fig. 1b).

The core of these six TMHs formed in yeast by Sdh3 and Sdh4 is similar to that in the human and avian structures, with RMSDs of 1.32 Å (182/200 matched residues) and 1.72 Å (188/200 matched residues), respectively. More structural details for Sdh3 and Sdh4 are provided in Supplementary text (Small subunits).

Larger differences are seen in the N-terminal extensions of yeast Sdh3 and Sdh4 in the matrix, and the distal end of the complex near the intermembrane space (IMS) (Figs. 1 and 3c, d and Supplementary Fig. 7).

The N-terminal α-helix of the yeast Sdh3, longer in yeast than in vertebrate structures (Figs. 3c and 4), packs against Sdh1, instead of the groove between Sdh1 and Sdh2, and is effectively shifted downwards with an angle of about 40 degrees compared to the human and avian structures (Fig. 3c). A long N-terminal extension of Sdh4, absent in vertebrates, occupies the position of the N-terminal SDHC helix in human CII and extends as a random coil up to the Q site (Figs. 1, 3a, c). Compared to vertebrates, these N-terminal extensions in yeast result in a greater number of stabilizing interactions, and a significantly larger interaction interface, between the membrane anchor and the hydrophilic catalytic subunits (Supplementary Fig. 9). Indeed, in the avian structure, the N-terminal α-helix of SDHC interacts with 12 residues of SDHB for an interaction interface of 553 Å². In yeast, the N-terminal coil of Sdh4 interacts with 14 residues of Sdh2 while the N-terminal α-helix of Sdh3 interacts with 21 residues of Sdh1, forming interaction interfaces of 765 Å² and 526 Å², respectively. These extended inter-domain interactions in yeast suggest extra support for the complex integrity, effectively linking more tightly the small membrane subunits to the hydrophilic catalytic domain (see also below).

Differences are also found at the distal end of the small subunits near the IMS. The two short α-helices, α3 from Sdh3 and the C-terminal α4' from Sdh4, are shifted about 4 Å deeper in the hydrophobic core compared to the vertebrate structures (Fig. 3d). Whether this is due to the absence of the heme group or to a thinner IMM, it forces the termini of the yeast TMHs into a more open conformation at the IMS side, with a difference of more than 9 Å between the distal ends of helices α4 of Sdh3 and α1' of Sdh4 compared to the avian structure (Fig. 3d). Furthermore, a conserved phospholipid at the IMS side is buried deeper into the hydrophobic core where it partially replaces hydrophobic heme interactions found in heme-containing complexes (Supplementary Fig. 8).

The CII Sdh3/Sdh4 assembly is similar to the one recently found in the cryo-EM structure of the yeast TIM22 translocation complex (PDB ID 6LO8) between Sdh3 and the paralog of Sdh4, Tim18[48]. The heterodimer superimposes well onto our yeast Sdh3 and Sdh4 (RMSD of 0.897 and

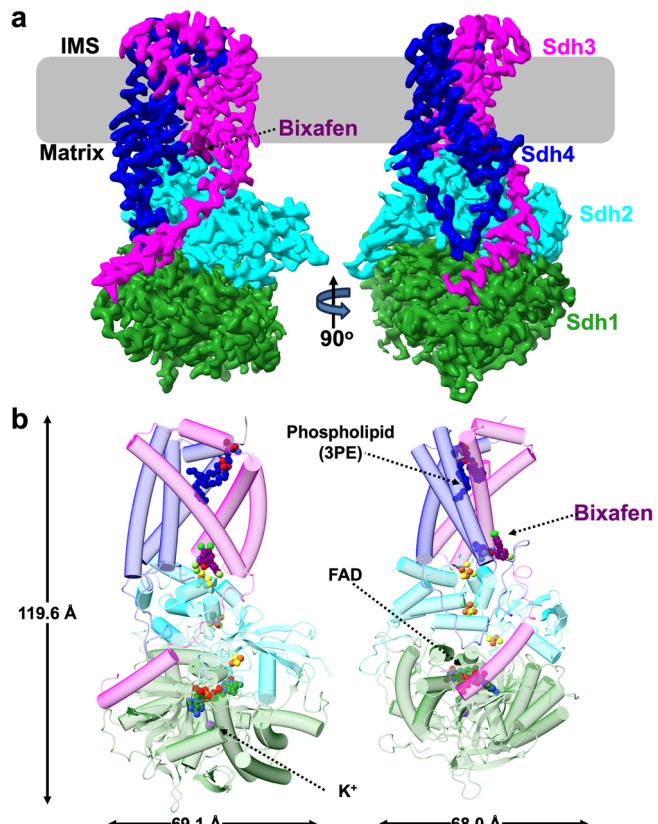

**Fig. 1 | Structure of the *S. cerevisiae* CII with the Q site inhibitor bixafen bound.** Two views of the complex rotated by 90 degrees are shown. **a** Cryo-EM map of CII at 3.00 Å resolution (threshold 0.52). Sdh1, Sdh2, Sdh3, Sdh4 and the inhibitor are colored in green, cyan, magenta, dark blue and purple, respectively. The membrane is represented as a grey area with the matrix and intermembrane space (IMS) sides indicated. **b** Cartoon representation of the CII model in similar orientations and protein color code as in (**a**). The FAD cofactor and K⁺ ion (in Sdh1), iron-sulfur clusters (in Sdh2, yellow and orange), bixafen (at the Q site, purple) and a phospholipid (dark blue) are shown as spheres. The dimensions of the complex are indicated.

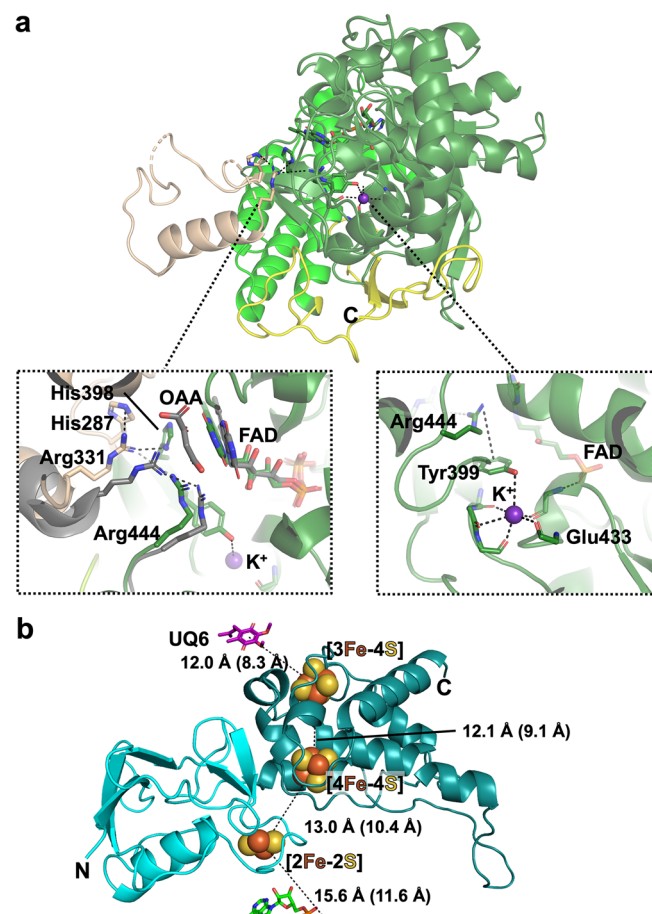

**Fig. 2 | The hydrophilic yeast CII subunits. a** The four domains of Sdh1 are represented in different colors: the FAD domain (residues 1–286 and 397–482) in dark green, the cap domain (residues 287–396 with dashed lines indicating non observed residues 331–356 and 391–396) in beige, the helical domain (residues 483–577) in bright green and the C-terminal domain (residues 578–635, labelled with C) in yellow. The left inset shows the superposition of the *S. cerevisiae* and avian (PDB ID 6MYO, grey) structures highlighting the relative positions of selected arginine residues and FAD cofactors, as well as the oxaloacetate (OAA) which is only present in the avian enzyme. The right inset displays the interactions of the K⁺ ion with the FAD and other key residues. **b** The Sdh2 protein is represented with its N-terminal (N) domain colored in cyan and its C-terminal domain (C) in darker (deep teal) shade. Center-to-center (and in parentheses edge-to-edge) distances between the redox centers are indicated.

1.008 Å), especially towards the IMS (Supplementary Fig. 10). While the Sdh3 sequence in the TIM22 complex is essentially the same as the Sdh3-W303 variant (i.e., Sdh3 of strain W303-1B) from our work, Tim18 displays significant sequence differences with Sdh4 (Fig. 4b) (see also below). Nevertheless, they both have a tyrosine (Tyr108 in Sdh4) in lieu of the heme-binding histidine found in e.g., vertebrate homologs, and the heterodimers they form with Sdh3 both lack heme cofactor. Differences at the N-termini of Sdh3 and Tim18 (mostly disordered and unresolved in the TIM22 structure) with Sdh3 and Sdh4 in our CII structure, may be related to the different nature of the greater complexes they form, with the hydrophilic domain of TIM22 protruding into the IMS while this of CII protrudes into the matrix (Supplementary Fig. 10).

### Absence of heme and complex integrity

An interesting aspect of the yeast CII structure is the absence of heme, in particular considering its overall similarity to the mammalian and other C-type forms of the enzyme, where the heme has been suggested to play a structural role, cross-linking the two small subunits and stabilizing the entire complex[17,49]. In the yeast structure, the absence of heme is compensated by four additional H-bonds between the two small subunits (Fig. 3a, b and Fig. 4, black/red stars), the replacement of key residues to increase hydrophobicity in the TMHs core (Fig. 4, black circles), and as mentioned above, additional interactions mediated by the conserved phospholipid.

The His-heme-His assembly is replaced in the yeast structure by a direct H-bond between His156$_{Sdh3}$ and Tyr108$_{Sdh4}$ (Fig. 3a, b and Supplementary Fig. 8). Cys109$_{Sdh4}$, which has been proposed to act as a heme ligand[50] doesn't appear to make any contribution to the complex integrity (Fig. 3b). The interactions made by the two heme propionate groups in the heme-containing structures, are replaced in *S. cerevisiae* by three additional direct H-bonds between Arg97$_{Sdh3}$-Glu112$_{Sdh4}$, Asp167$_{Sdh3}$-Lys73$_{Sdh4}$ and Tyr157$_{Sdh3}$-Thr84$_{Sdh4}$ (Fig. 3b). Importantly, all these stabilizing interactions involve yeast residues that are not conserved from heme-containing CII sequences; four are in Sdh4 (Tyr108, Glu112, Lys73, Thr84) and one in Sdh3 (Tyr157) (Fig. 4).

The stability of the small subunits structure is further supported by additional hydrophobic interactions. The yeast complex contains five phenylalanines at positions 107, 114, 149, 153 of Sdh3 and position 101 of Sdh4, that increase the hydrophobicity of the core of the structure. The avian and human complexes contain leucine or alanine residues at these positions (Fig. 4).

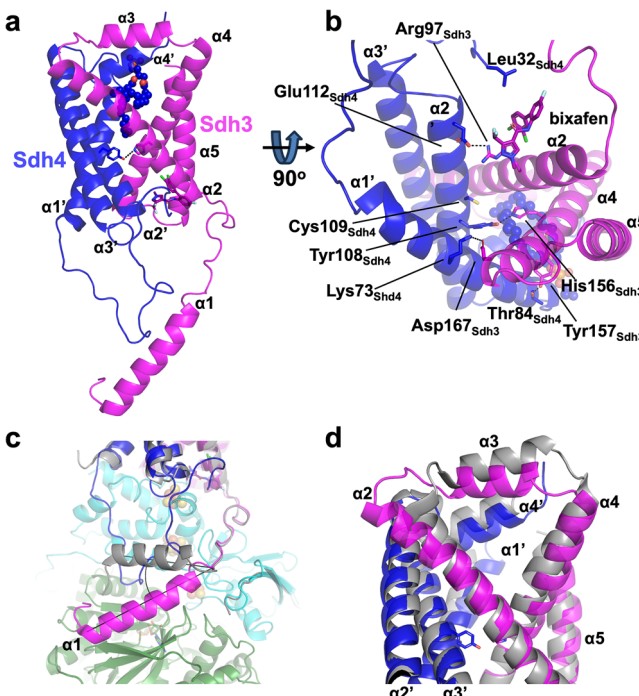

**Fig. 3 | Interactions of the hydrophobic CII proteins. a** Overall conformation of Sdh3 (magenta) and Sdh4 (dark blue) with phospholipid (spheres) and inhibitor bixafen (sticks) present. The central H-bond between His156$_{Sdh3}$ and Tyr108$_{Sdh4}$ is also shown. **b** 90-degree rotation from panel A highlighting the major H-bonds that stabilize the two proteins. **c** Superposition with the avian structure (PDB ID 6MYO, grey) highlighting the fungi-specific extensions at the N-terminals of Sdh4 (dark blue) and Sdh3 (magenta) and the down shift of Sdh3 N-terminal α-helix (α1) over Sdh1 (green). **d** Superposition of the *S. cerevisiae* (colored as in Fig. 1) and avian (grey) structures at the IMS side.

## Q site and inhibitor binding

The Q site of the yeast CII involves residues from Sdh2, Sdh3 and Sdh4. In the CII-nat structure, the Q site is occupied by an endogenous ubiquinone molecule (UQ6, Supplementary Fig. 2 and 3b). Due to a lack of continuous density, the modelled UQ6 molecule is structurally equivalent to ubiquinone-1, with the remaining of the tail disordered most likely in the lipid phase; Supplementary Fig. 2 and 3b). In the CII-bix structure, clear density in the cryo-EM map (Fig. 1 and Supplementary Fig. 3a) allowed us to build the bixafen molecule with high confidence. Despite the overall differences in their membrane domain, the yeast Q site (Figs. 4 and 5) is very similar to those of vertebrate, nematode and even *E. coli* CII[17,20,22]. Almost all of the residues of the Q site are conserved, and these residues are involved in binding ubiquinone in the same way that UQ8 is bound to the *E. coli* enzyme (PDB ID 1NEK)[17] or bixafen like other benzanilide-type SDH inhibitors such as flutolanil and carboxin to the avian (PDB IDs 6MYO, 2FBW)[21,37], porcine (PDB ID 4YXD)[38] and *Ascaris* (PDB ID 3VRB)[22] enzymes (Supplementary Fig. 11).

Focusing on the bixafen interactions, residues corresponding to Trp194$_{Sdh2}$ and Tyr120$_{Sdh4}$ have been considered universal ligands of the substrate and inhibitors, although the apparent bonds are often too long for strong H-bonds. In the yeast CII-bix structure there is a credible H-bond between Trp194$_{Sdh2}$ and the amide carbonyl oxygen of bixafen (3.0 Å), but the corresponding distance from that atom to the Tyr120$_{Sdh4}$ hydroxyl is rather a moderate electrostatic interaction (3.5 Å). Trp194$_{Sdh2}$ is also in π-stacking interaction with the central aromatic ring of bixafen (Fig. 5a, b). On the other side of the inhibitor, Ser94$_{Sdh3}$ forms a moderate interaction with the amide N atom (3.8 Å), further supported by hydrophobic interaction with Ile98$_{Sdh3}$, while no interaction is observed with Ser93$_{Sdh3}$, be it direct, as observed in the CII-nat structure with the UQ6 molecule (Supplementary Figs. 2b and 3c), or via a water molecule, as previously reported in other

structures[21,37]. The inhibitor is further stabilized by hydrophobic interactions involving residues Pro190 and Ile239 of Sdh2, Leu81 and Trp90 of Sdh3 and Tyr120 of Sdh4 (Fig. 5a, b).

Although there is no other available CII structure with bixafen bound, we can compare bixafen binding to other inhibitors in the same class resembling carboxanilides, such as flutolanil. Comparing with flutolanil binding in the avian structure (PDB ID 6MYO), the aromatic carboxylate groups of the two inhibitors superimpose closely, including the ortho di- or tri-fluoro-methyl groups and the amide linkage (Supplementary Fig. 11b). The smaller ring of bixafen partially compensates for its extra methyl group, which pokes into a spacious cavity between four residues: Ser93$_{Sdh3}$, Arg97$_{Sdh3}$, Ile232$_{Sdh2}$ and His237$_{Sdh2}$ (Fig. 5b). The aniline rings of the two inhibitors nearly superimpose, but with flutolanil slightly closer to the protein and its meta-isopropoxy substituent packing among three residues at the far-end of the pocket, while that of bixafen in yeast is tilted slightly away from the protein, with its ortho-dimethylphenyl group directed away from the protein, out of the pocket.

In both CII-bix and CII-nat, the guanidinium of Arg97$_{Sdh3}$ is in H-bond interaction with Tyr120$_{Sdh4}$ and forms a salt bridge with Glu112$_{Sdh4}$ (Figs. 5 and 3b). While Arg97$_{Sdh3}$ and Tyr120$_{Sdh4}$ are conserved in heme-containing SQRs, Glu112$_{Sdh4}$ replaces a glycine residue (Fig. 4) and its interaction with Arg97$_{Sdh3}$ substitutes this of a heme propionate, suggesting a role for this residue in catalysis or inhibitor binding when a heme co-factor is absent.

The major difference in the yeast CII Q site compared to any other known SQR structure is the N-terminal sequence of Sdh4. This is fully resolved in the cryo-EM map with the N-terminal Leu32$_{Sdh4}$ (the first amino-acid after the signaling peptide) forming a hydrophobic patch with Trp194$_{Sdh2}$, Tyr120$_{Sdh4}$, Leu81$_{Sdh3}$ and Ile98$_{Sdh3}$ which is entirely enclosing the linker and hydrophobic rest of the inhibitor (Fig. 5a, b). Interestingly, the first three residues of Sdh4 (Leu32, Thr33 and Ile34) are disordered in the CII-nat structure, with the position occupied by the side chain of Leu32 in the CII-bix structure occupied by the aliphatic chain of UQ6 (Fig. 5c). Our structures thus highlight a unique feature of the yeast CII – the N-terminus of Sdh4 – which is absent from structures of other organisms, and either interacts with the hydrophobic inhibitor or becomes disordered when the larger native ubiquinone-6 substrate is bound.

A key residue in the Q site is His237$_{Sdh2}$. It is located near but not within H-bonding range of the core of the inhibitor and makes a H-bond with Asp119$_{Sdh4}$ (Fig. 5a). The significance of this is not known, but mutation of either of these residues leads to carboxin resistance[35,51–56].

## Comparison with the AlphaFold models

The predicted AlphaFold protein models of the four yeast CII subunits (Q00711, P21801, P33421, P37298) align reasonably well with those determined here by cryo-EM with RMSD values between 0.42 and 0.63 Å after removal of the modeled signaling peptides (Supplementary Table 1 and Supplementary Fig. 12). Nevertheless, significant differences can be seen in the protein N-termini, as well as in several side chain conformations, particularly on subunit interfaces, that couldn't be predicted by AlphaFold, resulting in loss of information on critical stabilization interactions. Further details are given in Supplementary text.

## Molecular docking analysis of SDHIs in the Q site of CII

To better understand the molecular basis of inhibition at the Q site, we performed docking simulations focusing on some SDHI fungicides which are widely used in agriculture, namely boscalid (first marketed by BASF), pydiflumetofen (Syngenta) and isoflucypram (Bayer). Similar to bixafen, they have been designed on a carboxamide core, followed by a linker and hydrophobic rest or tail (Fig. 6)[32].

We used the CII-bix structure as the initial model, viewed in the Maestro 14.0 interface, and followed a similar protocol for all ligands of structure preparation, receptor grid generation, ligand preparation and molecular docking with glide.

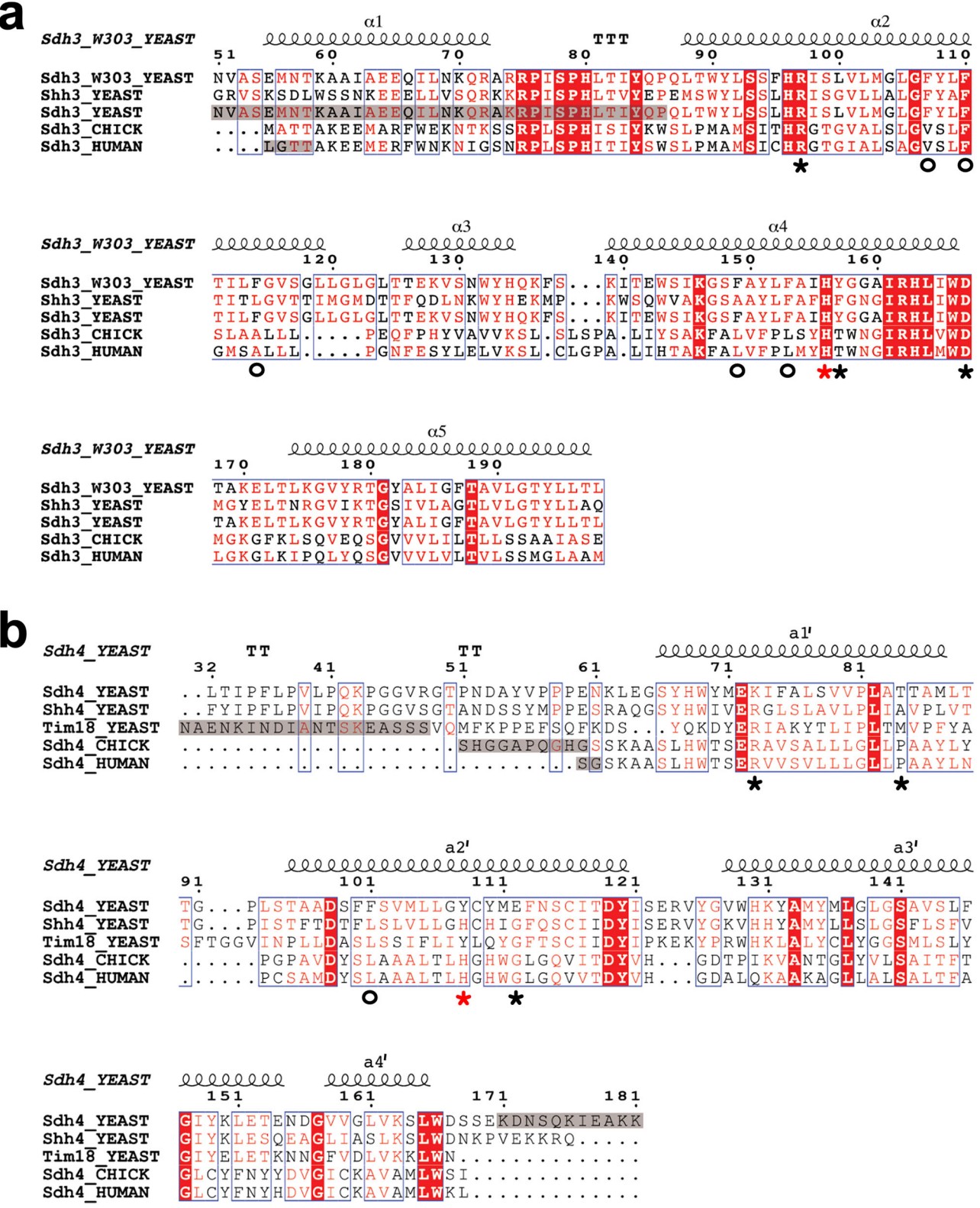

**Fig. 4 | Sequence alignment of Sdh3 and Sdh4 homologs.** Hybrid structure-based sequence alignment of Sdh3 (**a**) and Sdh4 (**b**) homologs from yeast, chicken and human. The secondary structure elements (helices, α and a', and turns, T) and numbering of the yeast CII determined in this work are shown above the aligned sequences. Alignment is based on the structures of the yeast complex presented here (Sdh3_W303_YEAST and Sdh4_YEAST, PDB ID 9QDM), the yeast Sdh3 and Tim18 of the TIM22 complex (PDB ID 6LO8), and the Sdh3 and Sdh4 homologs of the chicken (PDB ID 6MYO) and human (PDB ID 8GS8) CIIs. The sequence alignment also includes Shh3_YEAST and Shh4_YEAST, paralogs of Sdh3 and Sdh4, respectively, for which no structural information is available. Residues in red boxes are identical, residues in red are similar, blue frames show similarity across groups. Residues in grey background are disordered (not visible) in the structures. The red stars indicate the position of the histidines that ligate the heme (when a heme is present). The black stars and black circles mark the residues that in the yeast CII form stabilizing H-bond and hydrophobic interactions.

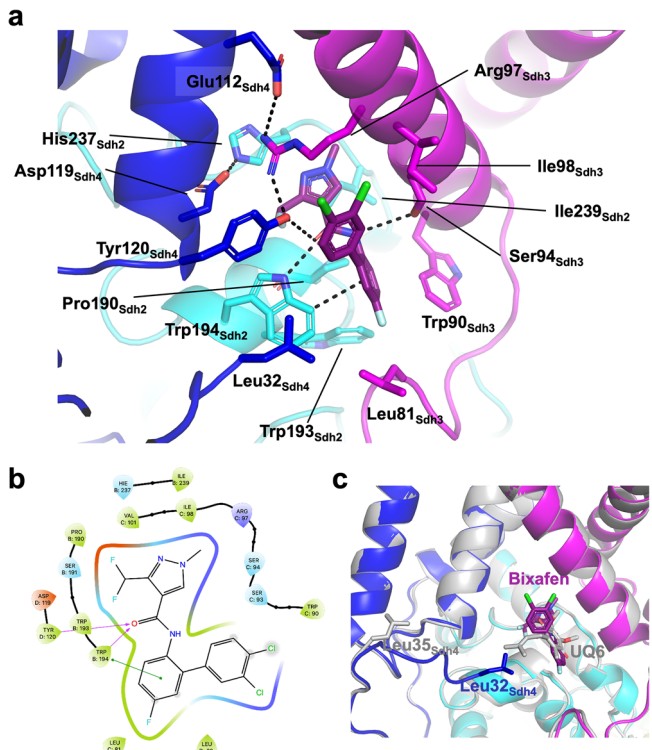

**Fig. 5 | Bixafen binding at the Q site of the yeast CII. a** Protein-inhibitor interactions. Dashed lines represent H-bonds and π-stacking interactions. The main contributing residues are labelled. CII proteins are colored as in Fig. 1 with bixafen in purple. **b** 2D stick diagram of bixafen in the active site. Pink arrows represent H-bonds and the green connector bar the π-stacking interaction. Red and blue colors indicate negative and positive charges interactions, light blue polar interactions and green hydrophobic interactions. Blue arrows indicate H-bonds and the red line π-π stacking interaction. The solvent exposed residues are grey shaded. **c** Superposition of the yeast CII-bix (in color) and CII-nat (grey) structures, with their first resolved Sdh4 residue labeled (Leu32 and Leu35, respectively).

The glide scores obtained were of −9.53 for boscalid, −8.74 for bixafen, −6.84 for isoflycypram and −5.60 for pydiflumetofen, suggesting a stronger interaction of boscalid and bixafen to the yeast enzyme (Supplementary Table 2).

The docking results were validated using the cryo-EM structure. Superposition of the CII-bix and glide generated bixafen coordinates resulted in a RMSD of 0.138 Å, signifying accuracy of the docking process.

Consistent with their conserved carboxamide core, docking of all inhibitor molecules resulted in a similar overall orientation although substitutions of the hydrogen ligand of the nitrogen carboxamide in isoflucypram and pydiflumetofen favored hydrophobic interactions on the amide side (Fig. 6). The main interactions of the inhibitors are therefore conserved and include polar interactions of the carboxyl group with $Trp194_{Sdh2}$ and $Tyr120_{Sdh4}$ and interactions of the amide group with Sdh3 and $Ser94_{Sdh3}$ in particular (polar for bixafen and boscalid; hydrophobic for isoflucypram and pydiflumetofen) (Fig. 6 and Supplementary Fig. 13). The pyrazole ring of bixafen, isoflucypram and pydiflumetofen as well as the pyridine ring of boscalid are all in the same position, forming mainly hydrophobic interactions with $His237_{Sdh2}$, $Ile239_{Sdh2}$, $Ile98_{Sdh3}$ and $Arg97_{Sdh3}$ (Fig. 6 and Supplementary Fig. 13). The remaining part of the inhibitors consists of a phenyl-phenyl group for bixafen and boscalid, an isopropyl-benzyl group for isoflucypram and a trichlorophenyl(ethyl)-1H-pyrazole for pydiflumetofen. All integrate a network of hydrophobic interactions mainly with $Try120_{Sdh4}$, $Leu32_{Sdh4}$ and $Trp90_{Sdh3}$ and π-stacking interactions with $Trp194_{Sdh2}$. The isopropyl-benzyl group of isoflucypram displays a similar orientation with its aromatic ring at an averaged position compared to this occupied by the phenyl-group of the other

two molecules, while the trichlorophenyl group of pydiflumetofen appears to be positioned closer to $Trp90_{Sdh3}$, although an alternative conformation is also possible given the chirality of the molecule (Fig. 6).

Overall, this simple docking analysis of selected fungicide molecules using our experimentally determined yeast structure, highlights the principles of their interactions, demonstrating the potential for in silico evaluation of the contribution of different substituents at different positions, in a fungal system.

## Similarity to pathogenic strains

As described above, a major difference between the yeast CII and previously determined structures from higher vertebrates or bacteria lies in the extended amino-acid stretch at the N-terminus of Sdh4 which appears to be a unique feature of the yeast sequences (Fig. 4). In *S. cerevisiae* this results in significant conformational changes in the inter-subunit interactions (Fig. 3c) as well as direct contribution at the Q site (Fig. 5 and Supplementary Fig. 2b). Comparison of the *S. cerevisiae* Sdh4 sequence with those of pathogenic fungi, selected for their importance in disease control (Supplementary Table 3), reveals striking similarities at their N-termini with several conserved or identical amino-acids (Fig. 7). Although there is no definitive information on the transit peptide length in these species, the conserved aligned residues and particularly the prolines in positions 41, 44, 58 and 60 (*S. cerevisiae* numbering) suggest a similar structural layout in these pathogenic fungi (Fig. 7). Furthermore, the Sdh4 sequence of *Candida glabrata* shows conservation with *S. cerevisiae* for residues Lys73, Thr84, Tyr108 and Glu112 which, as described above, is a very strong indication that *Candida glabrata* CII is also lacking heme cofactor in its membrane domain (Fig. 7).

All the residues from Sdh2, Sdh3 and Sdh4 that are seen to contribute to bixafen binding in *S. cerevisiae* (Fig. 5b) are conserved in these pathogenic strains (Supplementary Fig. 14a, b and Fig. 7a, diamonds), with a few exceptions that may confer sensitivity variations of their Q sites.

## Discussion

Because of the central role of respiratory CII in metabolism, inhibitors that target its Q site have evolved as one of the dominant classes of fungicides that are used globally in agriculture to control the proliferation of pathogens[32,57]. However, the lack of a fungal CII structure means that to date, no SDHIs have been developed based on clade-specific molecular details of the site they target, also limiting our understanding and ability to mitigate resistance mutations emerging in field isolates[58]. Our cryo-EM structure of the *S. cerevisiae* CII provides the first atomic details of a fungal enzyme and its Q site, in the presence of endogenous UQ6 or the SDHI bixafen. While it overall resembles the structure of known homologs, it also reveals some unique features, potentially signifying a pivotal point for the design of new SDHIs against pathogens that pose a critical threat to crops, animals, and humans.

The most significant structural feature of the *S. cerevisiae* CII is the extension of the Sdh4 N-terminus (residues 32–62) that induces major remodeling of the matrix interactions made by Sdh3 and Sdh4 with the catalytic subunits. Beyond supporting structural integrity of the entire complex, it also appears to be involved in binding at the Q site, interacting differently in our structures with the endogenous UQ6 and bixafen. Most interestingly, alignment of the Sdh4 sequence of *S. cerevisiae* with other yeast and non-yeast fungal homologs, indicates that this N-terminal extension of Sdh4 is a common feature of this clade (Fig. 7). Inspection of residues conservation across species suggests that their Sdh4 can adopt a similar fold in the matrix as seen in *S. cerevisiae* though this, and whether their tail end also interacts with molecules bound at the Q site, remain to be structurally established. SDHIs have shown large species specificity[38,59,60] and it is noteworthy that SDHIs like bixafen have been developed, and selected, to specifically target fungal pathogens[33,37]. It is therefore possible that their efficacy and specificity also arise from their thus far unexplored interaction with the N-terminal of Sdh4. Further work should be developed to specifically investigate the effect of sequence variation in this region, as well as in

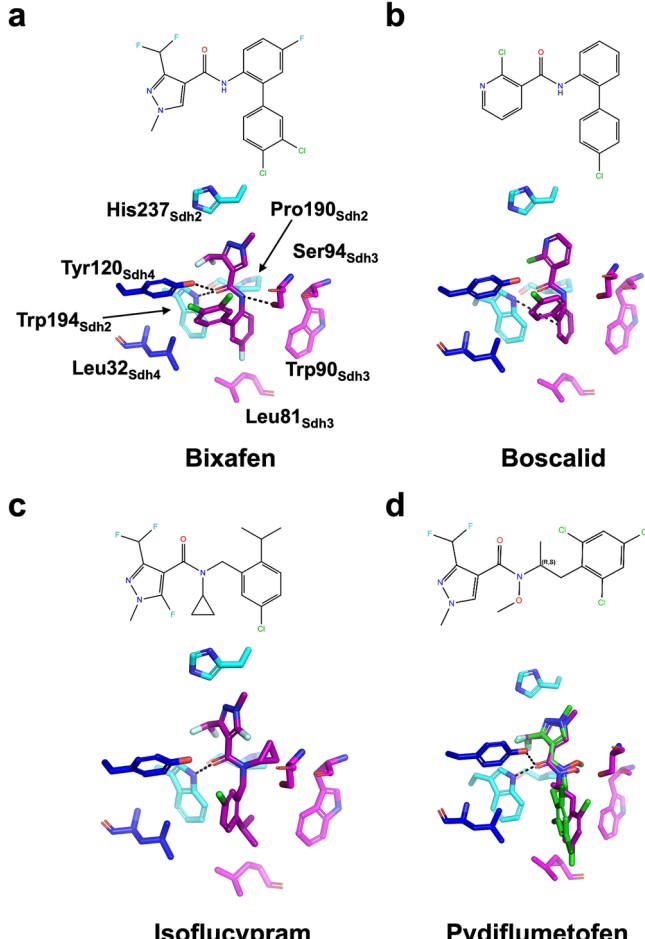

**Fig. 6 | Chemical structure and simulated binding of different FRAC group 7 fungicides at the Q site of the yeast CII. a** bixafen (pyrazole-4-carboxamide), (**b**) boscalid (pyridine-carboxamide), (**c**) isoflucypram (N-cyclopropyl-N-Benzyl-pyrazole-carboxamide), (**d**) pydiflumetofen (N-methoxy-(phenylethyl)-pyrazole-carboxamide). The molecular structure of each inhibitor is oriented with respect to their binding orientation, with the core interacting with His237$_{Sdh2}$ to the left followed by the carboxamide and solvent exposed hydrophobic rest to the right. The binding to CII was calculated in Maestro using Glide. The color coding of the CII subunits is as in Fig. 1. Calculated H-bond and π-stacking interactions are shown with dashed lines. The rest of the pydiflumetofen can adopt one of two conformers, R or S, represented in purple and green, respectively.

their Q site, on differences in the sensitivity of different fungi towards various inhibitors.

Another difference of *S. cerevisiae* CII, which may be a common feature of some yeasts but not all fungi, is the absence of a heme group within the membrane domain. Up until recently, when the first structures of divergent protozoan CIIs emerged[24–26], all eukaryotic CIIs had been classified into the same group C, characterized by the presence of one B-type heme in their membrane anchor domain. In the case of *S. cerevisiae*, the presence of a heme group has long been a matter of controversy which mostly remained unresolved because of the instability of the enzyme upon purification with only a brief report in 1982 of a successful isolation[61]. Our work clearly shows the absence of a heme co-factor in *S. cerevisiae* CII and calls for a subdivision of group C members to acknowledge the absence of heme in the membrane domain of at least thus far, *S. cerevisiae* and some protozoan enzymes.

Mutagenesis studies in *E. coli* SQR convincingly showed that the heme is not essential for enzyme assembly or redox activity and that its role may be to provide greater stabilization of the membrane anchor domain[43,62]. We

identified key structural features in *S. cerevisiae* that specifically stabilize Sdh3 and Sdh4 within the IMM including four H-bonds which replace the heme mediation and involve Sdh4 residues which are not conserved in known heme-containing homologs - namely Tyr108, Glu112, Lys73 and Thr84 - as well as additional hydrophobic interactions also mediated by the deeper burial of a conserved phospholipid (Figs. 3 and 4).

Of the Sdh4 pathogen sequences surveyed, *Candida glabrata* appears to also have evolved these key residues that in *S. cerevisiae* stabilize the hydrophobic core, suggesting its CII most likely also lacks heme co-factor. *C. glabrata*, recently renamed *Nakaseomyces glabratus*, is the second most isolated *Candida* species behind *C. albicans* in human infection, with a mortality of almost 50% in case of candidiasis in weakened immune system patients[63,64]. Consistent with analyses of ribosomal RNA sequences that suggest it is evolutionarily closer to *S. cerevisiae* than to other *Candida* species[65], *C. glabrata* also shares an almost identical Sdh4 N-terminal sequence with *S. cerevisiae* (Fig. 7b), indicating that our structure of the *S. cerevisiae* CII is a close model for that pathogenic yeast. The presence or absence of heme does also slightly modify the electrostatic network near the Q site (Glu112$_{Sdh4}$ – Fig. 7a) and could present an opportunity to differentially target pathogens depending on whether a heme cofactor is present or not. This is in addition to other subtle sequence variations seen in residues directly contributing to these pathogen Q sites (diamonds in Fig. 7a and Supplementary Fig. 14a, b).

It is notable that an ancient whole genome duplication in the *S. cerevisiae* lineage[66] generated multiple putative genes for CII subunits resulting in paralogues for all but the Sdh2 subunit. Sdh4 for instance has two paralogues, Shh4[67] and Tim18. Tim18 can associate with Sdh3 (or its paralog Shh3) but together cannot bind the hydrophilic domain to form a functioning CII isoform. Instead, the Tim18-Sdh3 heterodimer was found to be part of the inner membrane translocating complex TIM22[48]. Transformation into yeast of the other Sdh4 paralog, Shh4, showed that it could form a functional CII[68]. Curiously, Shh4 contains the canonical axial histidine and it could be that the assembled CII isozyme contains a heme B with question on the physiological relevance of the expression of different isozymes in response to changes in internal or external effectors[9]. Shh4 also has higher similarity of sequence and less phylogenetic distance to human SDHD than Sdh4[69]. It is possible it retains more ancestral features, and this could imply that the heme in *Saccharomyces* and other yeast species was lost after genome duplication.

Beyond structural support, the role of the B heme remains uncertain. Its position above the quinone binding site suggests that it is not involved in the catalytic electron transfer route but it has been suggested that it might delocalize the negative charge on the semiquinone intermediate, possibly increasing the forward catalytic rate[62], or be reduced by succinate when quinone is rate-limiting[70]. Both SQRs and QFRs are known to be reversible, but function more efficiently in their physiological direction. Reversibility of yeast CII has been demonstrated by AtpeninA-sensitive fumarate oxidation of B-type cytochromes in mitochondrial membranes[43]. Further experiments could be envisaged in yeast to express a heme-containing CII, possibly the Shh4 isozyme, to investigate differences in behavior with the wild-type heme-free enzyme and clarify the role of that redox co-factor.

This work completes our structural understanding of the electron transport chain and membrane embedded respiratory complexes of the yeast *S. cerevisiae*, highlighting novel features that may be characteristic of the fungal clade. It provides unprecedented molecular details of the fungal Q site and the binding of widely used SDHIs, which could prove instrumental in the development of new inhibitors that can tackle the problems of emergence of resistance but also toxicity[32,34]. *S. cerevisiae* also provides the opportunity of a genetically tractable system to develop future work and further our understanding of this central metabolic enzyme.

## Materials and methods
### Yeast strain and cell growth
The yeast strain used derived from W303-1B (MATα, leu2, trp1, ura3, ade2, his3), available from EUROSCARF, and harbors a 10-histidine tag preceded

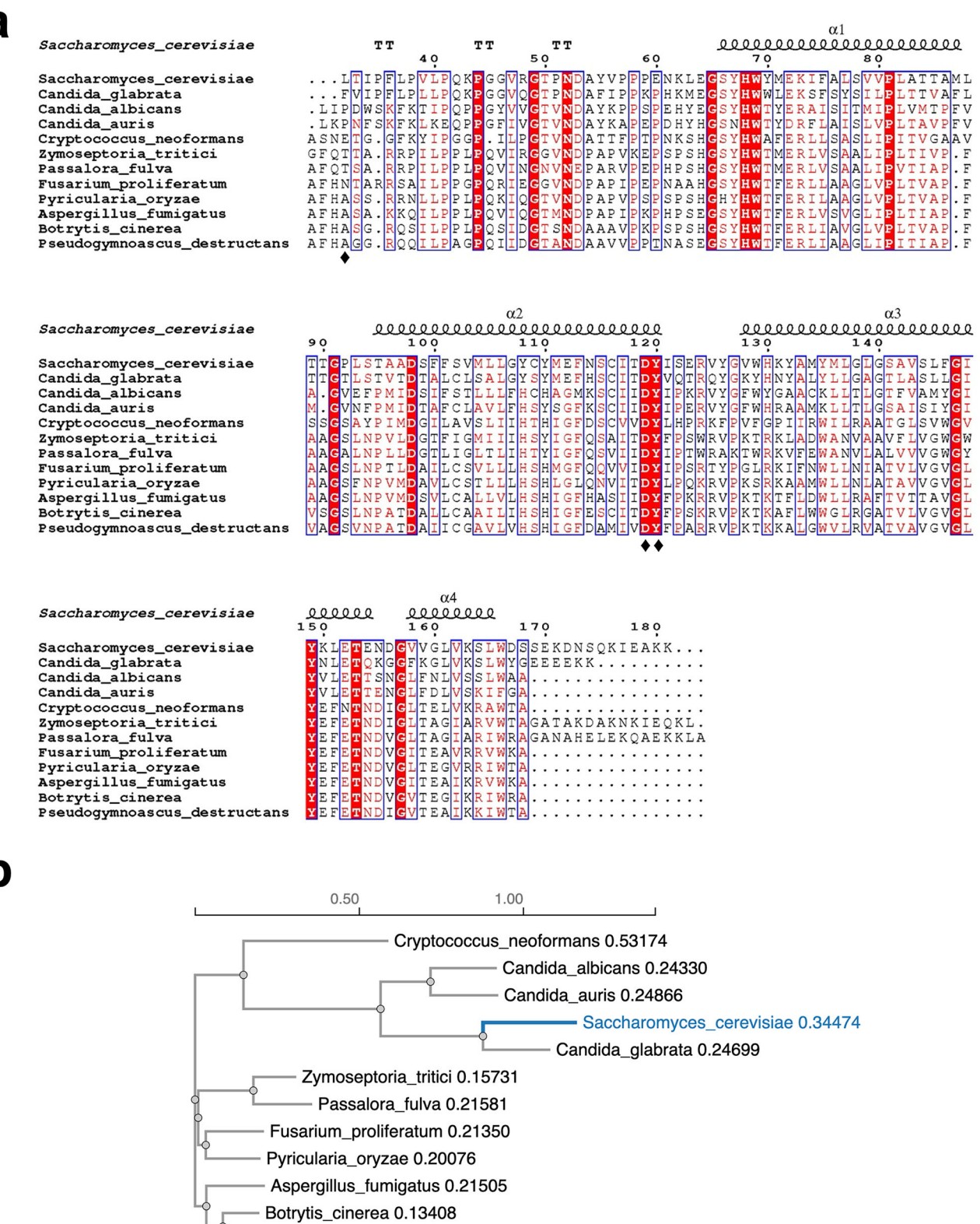

**Fig. 7 | Sequence alignment and phylogenetic tree of Sdh4 from *S. cerevisiae* and some pathogenic fungi. a** Sequence alignment. The secondary structure elements (helices, α, and turns, T) and numbering of the yeast CII determined in this work are shown above the aligned sequences. Residues in red boxes and white characters are identical, residues in red are similar, blue frames show similarity across sequences.

Residues that interact with bixafen in *S. cerevisiae* are indicated with diamonds. **b** Phylogenetic tree of the Sdh4 sequences with the position of the *S. cerevisiae* highlighted in blue. Branch lengths are distance corrected indicating evolutionary divergence, with a scale bar above the tree. The values indicate the extent of genetic change (higher numbers for larger extent of genetic change).

by a short linker (sequence GARGS) at the C-terminal of the Sdh4 subunit of CII. Cells were grown at 28 °C in 2 L baffled flasks containing 500 mL of YPEt medium (1% yeast extract, 2% peptone and 2% ethanol) with shaking at 200 rpm. Cells were harvested in late log phase by centrifugation at 9300 g for 5 min at 4 °C, and the resulting pellet was washed by two resuspension/centrifugation cycles (3600 g for 15 min at 4 °C) in 50 mM KPi pH 7.0. The final cell pellet was stored at -80 °C until use.

## Preparation of mitochondrial membranes

Thawed cells were resuspended in 50 mM KPi, 650 mM D-mannitol, 5 mM EDTA, and 0.1 mM PMSF at pH 7.4 and disrupted using glass beads (425 to 600 μm diameter) and a Bead-Beater® (Bio Spec Products Inc.) as described previously[71]. Cell debris were pelleted and discarded by centrifugation at 5600 g for 20 min at 4 °C, leaving a pink supernatant from which mitochondrial membranes were pelleted by centrifugation at 120,000 g for 50 min at 4 °C. The mitochondrial membranes were washed once by resuspension/centrifugation in 50 mM KPi, 100 mM KCl, 5 mM EDTA and 0.1 mM PMSF at pH 7.4, and twice in 50 mM KPi, 2 mM EDTA and 0.1 mM PMSF at pH 7.4. Mitochondria were resuspended in a minimal volume of 50 mM KPi, 2 mM EDTA and 0.1 mM PMSF at pH 7.4, aliquoted for repeat experiments and flash frozen in liquid nitrogen for storage at -80 °C until use. The total mitochondrial protein concentration was estimated using a Pierce™ bicinchoninic acid (BCA) protein assay (Thermo Scientific) according to the manufacturer's instructions.

## Complex II purification

Mitochondrial membranes were diluted to 2 mg/mL in 50 mM HEPES, 150 mM NaCl, and 1 mM PMSF at pH 8.0 before addition of 2% GDN (Anatrace). The solution was left to incubate for 1 h at 4 °C with gentle agitation. 350 mM NaCl and 10 mM imidazole were then added before centrifugation at 120,000 g for 30 min at 4 °C to remove insolubilized material before loading the supernatant onto a HisTrap HP 5 mL column (Cytiva) equilibrated with 3 column volumes (CVs) of 50 mM HEPES, 500 mM NaCl, 10 mM imidazole and 0.05% GDN at pH 8.0. The column was then washed with 5 CVs of 50 mM HEPES, 150 mM NaCl, 10 mM imidazole and 0.05% GDN at pH 8.0, and CII was eluted in that same buffer but with imidazole concentration increased to 500 mM. The eluate was concentrated with a 100 kDa MWCO centrifugal concentrator (Sartorius Vivaspin™) before loading onto a Superose 6 Increase 10/300 column (Cytiva) equilibrated with 50 mM HEPES, 150 mM NaCl, 0.05% GDN at pH 7.2 using an ÄKTA Pure™ Protein Purification System. The peak fraction was concentrated as described above and re-injected for a second gel filtration run to produce the final preparation.

## Analytical methods

The total protein and CII concentration in the final sample were estimated from a Pierce™ BCA assay (as above) and FAD content measured from a dithionite-reduced *minus* oxidized visible absorption spectrum, respectively, using for the latter a house-built spectrophotometer and molar extinction coefficient (ε) at 450 nm of 11.3 mM$^{-1}$ cm$^{-1}$ [72]. Clear-Native and Blue Native PAGE were performed with pre-cast 3–12% Bis-Tris NativePAGE gels (Invitrogen) as per the manufacturer instructions using the NativeMark™ Unstained Protein Standard (Invitrogen) for molecular weight estimation. In-gel activity assays for CII were performed using published protocols[73] and Clear-Native PAGE gel strips, with incubation time of 40 minutes.

## Activity assays

Spectrophotometric assays were conducted on a AvaSpec-ULS2048CL-EVO spectrophotometer at room temperature (22 °C) using the time series function at 600 nm (Avantes). The assay was carried out in a solution of 50 mM HEPES, 150 mM NaCl and 0.05% GDN at pH 7.2, supplemented with 80 μM decylubiquinone (DQ), 40 μM 2,6-dichloroindophenol (DCIP) and 5 mM succinate. The reaction was initiated by addition of 21, 42 or 84 μM CII and terminated by addition of 5 mM malonate. Turnover

numbers were calculated by linear regression using $\varepsilon_{DCIP}$ at 600 nm of 21 mM$^{-1}$.cm$^{-1}$ [74], subtracting the baseline malonate rate and expressed (x2) as an e.s$^{-1}$ rate. For inhibitor assays, bixafen was added during the linear phase of the reaction at concentrations of 0.5, 1 and 5 μM and rates were calculated as described above.

## Cryo-EM grid preparation and data acquisition

4 μL of freshly purified samples (at final protein concentrations of 25 μg/mL for the native sample and 21 μg/mL for the bixafen sample, allowing for the latter 1 h incubation on ice at a 2:1 bixafen:CII molar ratio) were applied to graphene oxide (Sigma-Aldrich, 763705-25 ML) coated C-flat TEM grids (Electron Microscopy Sciences, C-flat™, CF313-100, R 1.2/1.3, 300 copper mesh) and plunge-frozen in liquid ethane (−180 °C) using a EM GP2 Automatic Plunge Freezer (Leica Microsystems) with the following settings: sensor blotting with an additional move of 0.2–0.3 mm, 0 s delay time before blotting, front blotting with a blot time of 5 s, 6 °C and 90% humidity.

The data were collected at the ISMB Birkbeck EM facility using a Titan G3i Krios microscope (Thermo Fisher Scientific) operated at 300 keV and equipped with a BioQuantum energy filter (Gatan). The images were collected with a post-GIF K3® direct electron detector (Gatan), operated in super-resolution mode, at a magnification of 105,000 corresponding to a pixel size of 0.828 Å. The GIF slit was centered on the zero-loss peak, and the slit width was set to 20 eV. The dose rate was set to 13.6 e per pixel per second and a total dose of 49.5 e.Å$^{-2}$ was fractionated over 50 frames. Data were binned to physical pixel size before saving as LWZ compressed non-gain normalized TIFF movies. Data were collected using the EPU (version 3.4) software with a defocus range −2.7 μm to −1.2 μm. Approximately 20,000 movies were collected for each sample.

## Image processing

All steps in image processing for both datasets were performed in CryoSPARC[75] version 4.3. A total of 19,504 movies for the CII-nat and 19,804 for the CII-bix were subjected to patch motion correction and CTF estimation. After rejecting 112 and 167 micrographs for the CII-nat and CII-bix data sets, respectively, the remaining micrographs were used for particle picking setting a 130 Å particle diameter and a 20 Å lowpass filter for micrographs. Particles were extracted with a box size of 340 pixels. After several rounds of 2D classification 445,393 and 699,876 particles for the CII-nat and CII-bix datasets were selected and used for ab initio reconstruction with four classes each.

For CII-nat, particles corresponding to three out of these four reconstructions were merged and used for heterogenous 3D refinement using the same three ab initio reconstructions as starting models. Only one of the generated reconstructions was interpretable while the other two were highly anisotropic. Particles corresponding to this model were subjected to one more round of iterative 2D classification and selected particles were then used for non-uniform 3D refinement, generating the final high-resolution reconstruction at 3.15 Å according to the gold standard Fourier Shell Correlation (FSC) with 0.143 cutoff.

For CII-bix, one of the four ab initio reconstructions was of significantly higher quality and was used for two rounds of heterogeneous 3D refinement followed by non-uniform 3D refinement resulting in a final reconstruction with a resolution of 3.0 Å according to the gold standard FSC with 0.143 cutoff. The image processing workflow is summarized in Supplementary Fig. 4.

Local resolution varies between 2.7 and 4.2 Å for both maps. Angular distributions of particles indicate areas of higher particle populations consistent with earlier failed 3D reconstruction attempts due to preferred orientation (Supplementary Fig. 5).

## Model building

The cryo-EM maps were sharpened using the auto_sharpen tool in the Phenix suite[76] (v1.21.1-5286-000). The sharpened maps were used to fit the AlphaFold models AF-Q00711, AF-P21801, AF-P33421, and AF-P37298 corresponding to the Sdh1, Sdh2, Sdh3, and Sdh4 subunits of CII using

UCSF Chimera[77] (v1.19). Further, all prosthetic groups including the FAD, the iron-sulfur clusters, the lipids, the potassium ion and the ubiquinone-1 (CII-nat), bixafen (CII-bix) were fitted using Coot[78] (v. 0.9.8.96). The model was further improved through iterations of model building in Coot and real space refinement in Phenix. All proteins except part of the cap domain in Sdh1 and the C-terminus of Sdh4 fitted well into the cryo-EM map. Geometry definitions of the ligands were available in the Phenix ligand library apart from bixafen which was generated by phenix.elbow. For real space refinement, we used secondary structure restraints as well as Ramachandran and rotamer restraints. The potassium ion was further validated with the CheckMyMetal server[79], which confirmed that only potassium is compatible with this coordination site. Refinement and model statistics are summarized in Table 1. Maps and molecule representations in all figures were prepared using PyMOL v. 3.0.3 (https://www.pymol.org/) and UCSF Chimera.

## Computational modeling

The computational simulations were carried out using the Maestro portal v. 2024-2 (Shrödinger, Inc.). The CII-bix structure was prepared in the "protein preparation" wizard of Maestro by adding hydrogens, adjusting side chains to pH of 7.4, and optimizing the side chains and charges using the OPLS4 energy minimization protocol. All ligands (including bixafen as a reference) were prepared by LigPrep. Glide was used first to generate the receptor grid at the Q site and then to dock the ligands. For docking, the settings included extra precision (XP), aromatic hydrogens as H-bond donors, and halogens as H-bond acceptors. No other constraints on H-bonding were set. Binding was evaluated based on the Glide G score (Supplementary Table 2).

## Sequence alignments

All sequence alignments were performed with the Align tool in the Uniprot website (www.uniprot.org) and rendered with the ESPript server[80].

## Statistics and Reproducibility

Activity assay data were processed in Origin2023 using the Linear Fit function to calculate linear rates, then the means and standard deviations were calculated and plotted in Excel. Gel filtration profiles were consistent across four independent protein preparations. An initial 3.7 Å resolution EM model of CII was built using data collected on protein purified in a separate experiment from the final structures. Native gels showed a single band for purified CII in two independent protein preparations. FAD and total protein concentration calculated from reduced-minus-oxidized absorption spectra and NanoDrop A280 (mg/mL) readings were consistent within the same order of magnitude for two protein preparations.

## Reporting summary

Further information on research design is available in the Nature Portfolio Reporting Summary linked to this article.

## Data availability

E.M. maps and model coordinates for CII have been deposited in the EM Data Bank and Protein Data Bank, respectively, under accession numbers EM-53029 and PDB-9QDL for CII as purified (CII-nat), and accession numbers EM-53030 and PDB-9QDM for CII with bixafen bound (CII-bix). Raw data presented in Supplementary Fig. 1 for the characterization of the final purified CII sample used for structure determination are available in the Supplementary Data file, with uncropped gels provided in Supplementary Fig. 15.

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

## Acknowledgements

This work was supported by the Medical Research Council UK (Transition Support MR/T032154/1 to A.M.). Cryo-EM data were collected at the ISMB EM facility (Birkbeck College, University of London) with financial support from the Wellcome Trust (202679/Z/16/Z and 206166/Z/17/Z). We thank Dr D. Houldershaw for his support with computing.

## Author contributions

A.M. funded, designed, and supervised the research. B.M. produced the yeast mutant strain. C.B.-L. grew cells, prepared mitochondria, and purified complex II, with support from S.J. C.B.-L. performed activity measurements with input from E.A.B., S.C., and N.L., prepared and optimized cryo-EM grids, and collected cryo-EM data. N.P. and A.M. processed the cryo-EM images. N.P. and E.A.B. built the models. A.M., N.P., and E.A.B. wrote the manuscript with contributions from all authors.

## Competing interests

The authors declare no competing interests.
