## [Transparent Peer Review file · Communications Biology]

Cryo-EM structure of bixafen-bound *S. cerevisiae* complex II unravels SDHI specificity against pathogenic fungi

Corresponding Author: Professor Amandine Maréchal

Version 0:

Reviewer comments:

Reviewer #1

(Remarks to the Author)

This manuscript describes the first experimental structure of the respiratory chain complex II from the yeast *S. cerevisiae* in its native form, as well as bound to the inhibitor bixafen, a compound used against pathogenic fungi in agriculture.

The results are novel and of interest for the bioenergetic community, as well as for the agricultural industry and the conclusions are well justified by the experiments performed.

I believe this manuscript fits well with the scope and readership of Communications Biology, however I recommend a revision of the current version, to improve clarity and completeness.

Main comments:

-A supplementary figure showing the map quality for representative key features of the structures is missing. Usually an example of alpha helix and beta sheet, plus the binding sites for all cofactors are shown for each structure.

-Page 9: how do the authors assign potassium to the ion density in Sdh1? This should be discussed in the text.

-Page 13: the paragraph "Compared to vertebrates, these N-terminal extensions in yeast result in a greater number of H-bond stabilizing interactions between the membrane anchor and the hydrophilic catalytic subunits. The N-terminal α -helix of Sdh3 makes a majority of H-bonds (9 out of 12) with residues of Sdh1, while in the avian and human structures the majority of the SDHC H-bonds are with SDHB (11 out of 13). Interactions with Sdh2 are conserved but are now made with the extended N-terminal coil of Sdh4 which also interacts with the N-terminal α -helix of Sdh3." has no corresponding figure showing the different H-bond network. I suggest adding one.

-I would recommend adding a summary figure that shows the structural differences between vertebrate and yeast CII, highlighting the bound cofactors, beyond what is shown in Fig 1a and 3 (for example the heme binding site).

-Page 14: the sentence "In the yeast structure, the absence of heme is compensated by four additional H-bonds between the two small subunits, the replacement of key residues to increase hydrophobicity in the TMs core and, as mentioned above, additional interactions mediated by the conserved phospholipid." does not refer to a corresponding figure, I suggest adding one. In this regard, as Figure 3 is cited later in the description of the heme-binding residues, I suggest adding two panels similar to A and B depicting the vertebrate structure and highlighting the heme binding network, so it is easy to see the comparison with the yeast enzyme.

-On page 17: "The inhibitor is further stabilized by hydrophobic interactions involving residues Pro190 and Ile239 (not shown in figure) of Sdh2, Leu81 and Trp90 of Sdh3 and Tyr120 of Sdh4" all the results discussed should have a corresponding figure. I suggest the authors add one.

-Related to Figure 7, I suggest adding a supplementary figure showing the sequence alignment of the other SDH subunits between yeast and the mentioned pathogenic fungi. This is to provide better context to the docking performed in Figure 6 and show the conservation of the residues involved in inhibitor binding, mentioned in Fig 6. Since these include residues of Sdh2 and Sdh3, the current Figure 7 is not enough to assess the conservation of the binding site. This will enrich the discussion on pages 20-23.

Minor comments:

-Figure S1F-G: what error is depicted by the bars? There is no indication in the figure legend.

-Figures 1A, S3: the map threshold used for depiction should be reported.

-Figure 2A insets, 3A-B, S2B, 5A, 6: the length of the reported bonds is missing.

-Figure S8B: the consensus in the structural biology field is to color the best resolution in blue and the worst in red. I therefore recommend following this color scheme.

-Figure 2A: the four shades of green are not easily visible, I suggest using more different colors.

Reviewer #2

(Remarks to the Author)

This manuscript presents the first cryo-EM structures of *Saccharomyces cerevisiae* respiratory complex II with bound native ubiquinone-6 (modeled as UQ1) and with the fungicide bixafen. The key findings are as follows:

- High-resolution structures of fungal complex II in complex with both ubiquinone and the inhibitor bixafen.
- Absence of heme B in the membrane domain despite classification as a type C complex II.
- An extended N-terminal region in the membrane subunits Sdh3 and Sdh4 compared with mammalian enzymes, which may be relevant for the development of highly selective fungicides.

Overall, the work represents a significant advance in our understanding of the structure and function of fungal respiratory enzymes. I recommend publication after addressing the following points:

Major Comments

1. Modeling of ubiquinone binding

- a. The binding of native ubiquinone-6 (UQ6) in the native structure is modeled using short-chain ubiquinone-1 (UQ1). However, the overall binding mode of UQ and its differences from bixafen are not fully clear. Please provide the cryo-EM density for native UQ6 to clarify how UQ1 was modeled.
- b. In Supplementary Fig. 2, the prenyl chain appears to be oriented toward the hydrophilic domain; could the authors illustrate or discuss how full-length UQ6 is likely positioned?

2. Comparison of bixafen binding to other known inhibitor–complex II structures

- a. Bixafen (and other complex II inhibitors) have been structurally characterized in complex II from various sources (Refs. 21, 37, 38, 22). Please discuss how the binding mode observed in yeast differs from these previously reported structures. This will be of interest not only to specialists in respiratory enzymes but also to non-specialist readers.
- b. The authors note that the N-terminal region of the Sdh4 subunit may form a hydrophobic patch at the inhibitor (UQ) binding site (Lines 376–387). Please clarify whether this structural difference could contribute to intrinsic selectivity. If there are published reports on enzyme-level selectivity of complex II inhibitors between different species (e.g., animal vs. fungi), please refer to them.

Minor Comments

1. Lines 158–159: The authors state that the apparent molecular mass (~240 kDa) is twice the theoretical molecular weight of complex II, and that this can be expected for small membrane proteins. This statement is unclear. Please expand this explanation, referring to Ref. 44, so that it is understandable to a broad readership.
2. Lines 163–164: The IC_{50} value for bixafen (~0.5 μ M) is presented without context. Is this higher or lower than reported for other species? Inhibitor sensitivity is often used as a hallmark of the protein intactness. If the inhibition is sufficiently strong, please state that the purified enzyme is intact and catalytically active.
3. Figure 5c: The overlay of bixafen and UQ1 is small and difficult to interpret. Please enlarge this panel or provide a zoomed-in view to allow a clearer comparison of their binding modes.

Reviewer #3

(Remarks to the Author)

The manuscript “CryoEM structure of the bixafen inhibited *S. cerevisiae* respiratory complex II reveals the molecular basis of SDHI specificity against pathogenic fungi” describes a cryo-EM study of yeast respiratory complex II. The work presents the first experimental structure of CII in yeast, both active and inhibited with the fungicide bixafen. The study reveals the molecular details of the complex including the absence of heme b. The binding mode of bixafen is also established. Overall, this is a well-written and clearly illustrated manuscript, improving our knowledge on the details of CII in yeast and binding of inhibitors to the complex. I have only a few minor comments:

Page 6 and 16: It is stated that the Q site is partially occupied by endogenous ubiquinone in the native structure. Please clarify how it was established that there is partial occupancy. Is the density of poorer quality compared to the occupancy of the inhibitor in the bixafen structure? It would be beneficial to add the coulomb potential density map of UQ1 from the native structure in Supplementary Fig. 3 for comparison.

Page 14, 16 and Supplementary Information: Reference to Supplementary Information Fig. 3 follows after reference to Supplementary Information Fig. 4 in the text. For better readability, please switch the numbers on the two figures.

Version 1:

Reviewer comments:

Reviewer #1

(Remarks to the Author)

I am satisfied with the changes that have been made and recommend the manuscript in its current form for publication.

Reviewer #2

(Remarks to the Author)

The revised manuscript has successfully addressed my concerns.

Please extend my congratulations to the authors.

Reviewer #3

(Remarks to the Author)

The authors had addressed my comments in a satisfactory manner. I have no further concerns.

Responses to Reviewer Comments:

We would like to thank the three reviewers for their careful reading of the manuscript and the constructive comments made. We have provided below a detailed answer to each point raised and have modified the manuscript to address your concerns. To facilitate the review process, we have submitted 'track changes' versions of the main manuscript and SI files. We hope that you will find our answers satisfactory.

Kind regards,

Amandine

Referee expertise:

Referee #1: Structural biology, Mitochondrial Physiology, Biochemistry

Referee #2: Bioenergetics, Mitochondria, Chemical biology

Referee #3: Structure and functions of respiratory super complexes, Electron transfer

Reviewer #1 (Remarks to the Author):

This manuscript describes the first experimental structure of the respiratory chain complex II from the yeast *S. cerevisiae* in its native form, as well as bound to the inhibitor bixafen, a compound used against pathogenic fungi in agriculture.

The results are novel and of interest for the bioenergetic community, as well as for the agricultural industry and the conclusions are well justified by the experiments performed. I believe this manuscript fits well with the scope and readership of Communications Biology, however I recommend a revision of the current version, to improve clarity and completeness.

Main comments:

-A supplementary figure showing the map quality for representative key features of the structures is missing. Usually an example of alpha helix and beta sheet, plus the binding sites for all cofactors are shown for each structure.

We have added an additional Supplementary Figure (Supplementary Fig. 6) that provides examples of the map quality for representative key features including the FAD co-factor and potassium ion in Sdh1, the iron-sulfur clusters in Sdh2 and the bound lipid in the transmembrane region of the complex. The map for the ubiquinone-6 molecule was added to Supplementary Figure 3 as panel b, next to the bixafen map.

-Page 9: how do the authors assign potassium to the ion density in Sdh1? This should be discussed in the text.

The density was modeled as K⁺ based on ligand bond lengths, coordination geometry and comparison with analogous assignments in avian complex II structures.

X-ray studies have shown that the distance between a potassium ion and a carbonyl oxygen in proteins is reportedly longer (2.74 Å) than for e.g. sodium (2.38 Å) (please see Hardin 2006, a new reference added as number 47 in the main manuscript).

An ion at the same site in the avian complex II was modeled as K⁺ based on bond lengths and also on the strength of the anomalous signal (Huang et al. 2006).

Furthermore, we validated the position and identity of the metal using the CheckMyMetal server (Zheng H, Chordia MD, Cooper DR, Chruszcz M, Müller P, Sheldrick GM, Minor W (2014) Nature Protocols, 9(1), 156-70 - new reference added as number 80 in the main manuscript). This analysis clearly supported potassium as the only plausible ion at this position.

We have modified the main text to explain the basis for our density assignment as follows:

‘In the proximity of the succinate/fumarate binding site, our EM maps show density for a metal which appears to have a stabilizing role (Supplementary Fig. 6). We modelled a potassium ion based on ligand geometry and bond lengths⁴⁷ (Supplementary text, Flavoprotein Sdh1) as well as analogous modelling in the structure of the avian homologue²¹.’

with additional information of the modelling in the Model Building section of Materials and Methods:

‘The potassium ion was further validated with the CheckMyMetal server⁸⁰, which confirmed that only potassium is compatible with this coordination site.’

The metal-ligand bond lengths are now provided in Supplementary text, Flavoprotein Sdh1, as follows:

‘There is a metal-binding site adjacent to the dicarboxylate site with roughly octahedral ligation from five backbone carbonyl oxygens and the hydroxyl oxygen of Tyr399. The distances measured between the metal and the carbonyl oxygens (2.71 Å for Glu433, 2.79 Å for Gly402, 2.95 Å for Ala435, 3.07 Å for Asn400 and 3.46 Å for Met401) are consistent with the presence of a potassium ion for which longer distances are expected compared to e.g. sodium, magnesium or calcium¹.’

-Page 13: the paragraph “Compared to vertebrates, these N-terminal extensions in yeast result in a greater number of H-bond stabilizing interactions between the membrane anchor and the hydrophilic catalytic subunits. The N-terminal α -helix of Sdh3 makes a majority of H-bonds (9 out of 12) with residues of Sdh1, while in the avian and human structures the majority of the SDHC H-bonds are with SDHB (11 out of 13). Interactions with Sdh2 are conserved but are now made with the extended N-terminal coil of Sdh4 which also interacts with the N-terminal α -helix of Sdh3.” has no corresponding figure showing the different H-bond network. I suggest adding one.

The purpose of this paragraph is to describe the differences in the interaction interfaces between the subunits of the yeast and vertebrate complexes. Although the H-bond network provides a qualitative estimate of these interactions, it is not sufficient, and a figure showing

the H-bonds is hard to interpret as these are buried between the subunits and so are difficult to visualise.

To address this point, we have added a new figure in supplementary information (supplementary figure 9) that represents as surfaces the residues of the interaction interfaces made by the N-terminal extensions of the membrane subunits onto the hydrophilic subunits of the yeast and avian enzymes (two panels for the yeast structure highlighting interaction interfaces on Sdh1 and Sdh2 as they interact with Sdh3 and Sdh4, respectively, and one panel for the avian structure to show the interaction surface on SDHB as it is stabilized by SDHC). We have rewritten that paragraph as follows:

'Compared to vertebrates, these N-terminal extensions in yeast result in a greater number of stabilizing interactions, and a significantly larger interaction interface, between the membrane anchor and the hydrophilic catalytic subunits (Supplementary Fig. 9). Indeed, in the avian structure, the N-terminal α -helix of SDHC interacts with 12 residues of SDHB for an interaction interface of 553 \AA^2 . In yeast, the N-terminal coil of Sdh4 interacts with 14 residues of Sdh2 while the N-terminal α -helix of Sdh3 interacts with 21 residues of Sdh1, forming interaction interfaces of 765 \AA^2 and 526 \AA^2 , respectively..'

-I would recommend adding a summary figure that shows the structural differences between vertebrate and yeast CII, highlighting the bound cofactors, beyond what is shown in Fig 1a and 3 (for example the heme binding site).

As recommended, we have added supplementary figure 8 to highlight the major similarities and differences between the yeast and vertebrate (avian) CII structures to complement what is shown in other figures, notably Figure 3c and 3d.

-Page 14: the sentence "In the yeast structure, the absence of heme is compensated by four additional H-bonds between the two small subunits, the replacement of key residues to increase hydrophobicity in the TMHs core and, as mentioned above, additional interactions mediated by the conserved phospholipid." does not refer to a corresponding figure, I suggest adding one. In this regard, as Figure 3 is cited later in the description of the heme-binding residues, I suggest adding two panels similar to A and B depicting the vertebrate structure and highlighting the heme binding network, so it is easy to see the comparison with the yeast enzyme.

We have amended the text to point to the figures that highlight these interactions as follows:

'In the yeast structure, the absence of heme is compensated by four additional H-bonds between the two small subunits (Fig. 3a,b and Fig. 4, black/red stars), the replacement of key residues to increase hydrophobicity in the TMHs core (Fig. 4, black circles) and, as mentioned above, additional interactions mediated by the conserved phospholipid.'

Of note, this was an introductory sentence and these interactions are described in greater details (with reference to Fig. 3 and 4) in the paragraph that follows.

The new Supplementary Figure 8 shows the difference with the vertebrate CII structure and in particular the absence of heme in yeast. The focus of our work is the yeast Complex II,

and both Figure 3 and, in particular, the sequence alignment in Figure 4b, together illustrate these major differences relative to heme-containing structures (represented in Supplementary Figure 8).

-On page 17: "The inhibitor is further stabilized by hydrophobic interactions involving residues Pro190 and Ile239 (not shown in figure) of Sdh2, Leu81 and Trp90 of Sdh3 and Tyr120 of Sdh4" all the results discussed should have a corresponding figure. I suggest the authors add one.

This is an error in the text as all those residues are shown in Figure 5.

We have modified the text as follows to refer to Figure 5:

'The inhibitor is further stabilized by hydrophobic interactions involving residues Pro190 and Ile239 of Sdh2, Leu81 and Trp90 of Sdh3 and Tyr120 of Sdh4 (Fig. 5a,b).'

-Related to Figure 7, I suggest adding a supplementary figure showing the sequence alignment of the other SDH subunits between yeast and the mentioned pathogenic fungi. This is to provide better context to the docking performed in Figure 6 and show the conservation of the residues involved in inhibitor binding, mentioned in Fig 6. Since these include residues of Sdh2 and Sdh3, the current Figure 7 is not enough to assess the conservation of the binding site. This will enrich the discussion on pages 20-23.

We agree that highlighting the sequence conservation of the Q site will provide a better and wider context of inhibitor binding.

We have added a supplementary figure (Supplementary Figure 14) showing sequence alignments for Sdh2 and Sdh3 of *S. cerevisiae* and the pathogenic fungi selected for the alignment presented in figure 7a.

In both Figure 7a and Supplementary Figure 14, we show the residues involved in inhibitor (bixafen) binding (diamonds) as per Figure 5b.

We added at the end of the results the following description:

'All the residues from Sdh2, Sdh3 and Sdh4 that are seen to contribute to bixafen binding in *S. cerevisiae* (Fig. 5b) are conserved in these pathogenic strains (Supplementary Fig. 14a, b and Fig. 7a, diamonds), with a few exceptions that may confer sensitivity variations of their Q sites.'

Minor comments:

-Figure S1F-G: what error is depicted by the bars? There is no indication in the figure legend.

This is a clear an omission at submission stage and has now been addressed in the figure legend.

We also added a 'Statistics and Reproducibility' statement at the end of the manuscript as follows:

'Activity assay data were processed in Origin2023 using the Linear Fit function to calculate linear rates, then the means and SD were calculated and plotted in Excel. Gel filtration profiles were consistent across four independent protein preparations. An initial low resolution EM map was built using data collected on protein purified in a separate experiment from the final structures. Native gels showed a single band for purified CII in two independent protein preparations. FAD and total protein concentration calculated from reduced-minus-oxidised absorption spectra and NanoDrop A280 (mg/mL) readings were consistent within the same order of magnitude for two protein preparations.'

-Figures 1A, S3: the map threshold used for depiction should be reported.

This has been added to the figure legends of Fig. 1a and directly displayed onto supplementary figures 3 and 6 as requested.

-Figure 2A insets, 3A-B, S2B, 5A, 6: the length of the reported bonds is missing.

We avoided reporting bond lengths directly into figures, as this compromises clarity for readers. For cases that require particular attention we provide bond lengths (e.g. the potassium coordination) or commentary (e.g., credible hydrogen bonds, salt bridges, π -stacking interactions, or moderate electrostatic contacts) directly in the text.

-Figure S8B: the consensus in the structural biology field is to color the best resolution in blue and the worst in red. I therefore recommend following this color scheme.

Thanks for pointing that to us. We admit that we used the default options in the used software (Chimera, cryoSPARC and RELION) and have been consistently doing so in our previous works. Although recommended by this reviewer, we believe that the legend of the scale is clear and that no ambiguity can arise from the interpretation of the chosen colour scheme.

-Figure 2A: the four shades of green are not easily visible, I suggest using more different colors.

One of our co-authors is colour blind and these were the only distinguishable shades of green. We have slightly changed the colour scheme and we hope it is now easier to read.

Reviewer #2 (Remarks to the Author):

This manuscript presents the first cryo-EM structures of *Saccharomyces cerevisiae* respiratory complex II with bound native ubiquinone-6 (modeled as UQ1) and with the fungicide bixafen. The key findings are as follows:

- High-resolution structures of fungal complex II in complex with both ubiquinone and the inhibitor bixafen.

- Absence of heme B in the membrane domain despite classification as a type C complex II.
- An extended N-terminal region in the membrane subunits Sdh3 and Sdh4 compared with mammalian enzymes, which may be relevant for the development of highly selective fungicides.

Overall, the work represents a significant advance in our understanding of the structure and function of fungal respiratory enzymes. I recommend publication after addressing the following points:

Major Comments

1. Modeling of ubiquinone binding

a. The binding of native ubiquinone-6 (UQ6) in the native structure is modeled using short-chain ubiquinone-1 (UQ1). However, the overall binding mode of UQ and its differences from bixafen are not fully clear. Please provide the cryo-EM density for native UQ6 to clarify how UQ1 was modeled.

We have changed reference to UQ1 in our figures, text and are amending the pdb deposition to avoid confusion with structures where exogenous ubiquinone was added to the sample for structure determination and reinforce the fact that we assigned the density to endogenous ubiquinone-6 (UQ6).

That being said, and in response to this reviewer, the atoms in the tail were not visible in the density and so were not modelled.

We have now clarified that in the manuscript with the following statement:

'Due to a lack of continuous density, the modelled UQ6 molecule is structurally equivalent to ubiquinone-1, with the remaining of the tail disordered most likely in the lipid phase; Supplementary Fig. 2 and 3b)

and we now provide the cryo-EM density of UQ6 in Supplementary Fig. 3b.

Supplementary Fig. 2b gives a detailed view of UQ6 at the Q site of the yeast CII which binds, as written in the text, 'in the same way that [endogenous ubiquinone] UQ8 is bound to the *E. coli* enzyme' (PDB ID 1NEK, therein modelled as UQ2). It can be compared to bixafen binding in Fig. 5a.

Fig. 5c presents an overlay of bixafen and UQ6. However, the purpose of that figure is not to provide a detailed comparison of their respective binding modes. To try and address this, we have added an additional panel c in supplementary figure 3 which provides a different view of UQ6 binding (highlighting the residues involved in UQ6 binding) with bixafen from the CII-bix structure overlaid.

b. In Supplementary Fig. 2, the prenyl chain appears to be oriented toward the hydrophilic domain; could the authors illustrate or discuss how full-length UQ6 is likely positioned?

Although there are additional patches of density near UQ6 any interpretation could only be speculative and misleading given that the N-terminus of Sdh4 is also disordered compared to the bixafen inhibited CII structure. Most likely the UQ6 tail is free in the lipid phase and

disordered, and this is now commented in the text (please see additional text to the main manuscript in response to the previous comment above).

2. Comparison of bixafen binding to other known inhibitor–complex II structures

a. Bixafen (and other complex II inhibitors) have been structurally characterized in complex II from various sources (Refs. 21, 37, 38, 22). Please discuss how the binding mode observed in yeast differs from these previously reported structures. This will be of interest not only to specialists in respiratory enzymes but also to non-specialist readers.

There is no previous structure of bixafen bound to complex II in the protein database, but the binding mode is very similar to that of related fungicides.

We have added the following paragraph the inhibitor binding section:

‘Although there is no other available CII structure with bound bixafen, we can compare bixafen binding to other inhibitors in the same class resembling carboxanilides, such as flutolanil. Comparing with flutolanil binding in the avian structure (PDB ID 6MYO), the aromatic carboxylate groups of the two inhibitors superimpose closely, including the ortho di- or tri-fluoro-methyl groups and the amide linkage (Supplementary Fig. 11). The smaller ring of bixafen partially compensates for its extra methyl group, which pokes into a spacious cavity between four residues: Ser93_{Sdh3}, Arg97_{Sdh3}, Ile232_{Sdh2} and His237_{Sdh2} (Fig. 5b). The aniline rings of the two inhibitors nearly superimpose, but with flutolanil slightly closer to the protein and its meta-isopropoxy substituent packing among three residues at the far-end of the pocket, while that of bixafen in yeast is tilted slightly away from the protein, with its ortho-dimethylphenyl group directed away from the protein, out of the pocket.

and a new Supplementary Figure 11 comparing binding of bixafen in the yeast structure with flutolanil in the avian structure.

b. The authors note that the N-terminal region of the Sdh4 subunit may form a hydrophobic patch at the inhibitor (UQ) binding site (Lines 376–387). Please clarify whether this structural difference could contribute to intrinsic selectivity. If there are published reports on enzyme-level selectivity of complex II inhibitors between different species (e.g., animal vs. fungi), please refer to them.

We speculate that this may be the case however, we don't really have data in support except for the observation, in the discussion section, that bixafen (as other inhibitors tested in our docking) emerged on markets due to their selectivity towards fungi CII.

Inhibitor binding data for bixafen and other recently developed fungicides are not very available. Studies such as refs 32 and 57 (Scalliet, Fraaije) tend to report EC₅₀ values for the effective concentration against fungi- but this includes uptake, detox mechanisms as well as K_i, and they tend to compare different mutant strains, rather than different species.

Other SDHI inhibitors have shown large species specificity. For example flutolanil has been considered highly specific for *Ascaris*, with IC₅₀ values of 0.058 and 45.9 μM for *A. suum* QFR and porcine SQR (ref 38). Atpenin A5 inhibits CII with IC₅₀ of 32 nM for *Ascaris* vs 3.6, 3.7 nM for beef, rat (new ref 59). Finally, *E. coli* is highly resistant to carboxin compared to

mitochondrial CII (new ref 60). So it would not be surprising if involvement of the Sdh4 N-terminus affects specificity. We have slightly rewritten that part in the discussion as follows:

'SDHIs have shown large species specificity^{38,59,60} and it is noteworthy that SDHIs like bixafen have been developed, and selected, to specifically target fungal pathogens^{33,61}. It is therefore possible that their efficacy and specificity also arise from their thus far unexplored interaction with the N-terminal of Sdh4. Further work should be developed to specifically investigate the effect of sequence variation in this region, as well as in their Q site, on differences in the sensitivity of different fungi towards various inhibitors.'

Minor Comments

1. Lines 158–159: The authors state that the apparent molecular mass (~240 kDa) is twice the theoretical molecular weight of complex II, and that this can be expected for small membrane proteins. This statement is unclear. Please expand this explanation, referring to Ref. 44, so that it is understandable to a broad readership.

We expanded the statement as follows:

'This is greater than CII's molecular weight of 128 kDa and can be consistent with a monomeric form of the enzyme as the migration of small membrane proteins in native gels is strongly influenced by the lipid and detergent molecules they carry⁴⁴.'

2. Lines 163–164: The IC₅₀ value for bixafen (~0.5 μM) is presented without context. Is this higher or lower than reported for other species? Inhibitor sensitivity is often used as a hallmark of the protein intactness. If the inhibition is sufficiently strong, please state that the purified enzyme is intact and catalytically active.

As mentioned above, these values are not readily accessible (EC50 being more available) but we found reports of IC50 ranging from between 0.03-.1uM in *Z. tritici* field isolates and 2.50uM in *R. solani* and have added a reference to these for context.

We have added to this results section the following:

'This is within the range of IC50 values reported in the literature for bixafen on other fungi CII, from 0.03-0.1 μM for *Zymoseptoria tritici*⁴⁵ to 2.95 μM for *Rhizoctonia solani*⁴⁶. Taken together, these results support the intactness and functional integrity of our yeast CII preparation.'

3. Figure 5c: The overlay of bixafen and UQ1 is small and difficult to interpret. Please enlarge this panel or provide a zoomed-in view to allow a clearer comparison of their binding modes.

The main purpose of panel 5c is to emphasize the different behaviour of the Sdh4 C-terminus (ordered/disordered) upon ligand binding. This panel is also partially illustrating the distinct positions of bixafen and UQ6 relative to each other. Unfortunately, since bixafen is not a "flat" molecule, any overlay figure which will not provide any clear and interpretable

comparison with UQ6. For this reason, we chose the solution of the supplementary Figure 2b, where the position of UQ6 is clearly indicated relative to the key binding residues Tyr120(Sdh4), Trp194(Sdh2) and Trp90(Sdh3) which are also important for bixafen binding. We have now also added another view of UQ6 binding with an overlay of bixafen as panel c in supplementary figure 3 which we hope provide a clearer comparison.

Reviewer #3 (Remarks to the Author):

The manuscript “CryoEM structure of the bixafen inhibited *S. cerevisiae* respiratory complex II reveals the molecular basis of SDHI specificity against pathogenic fungi” describes a cryo-EM study of yeast respiratory complex II. The work presents the first experimental structure of CII in yeast, both active and inhibited with the fungicide bixafen. The study reveals the molecular details of the complex including the absence of heme b. The binding mode of bixafen is also established. Overall, this is a well-written and clearly illustrated manuscript, improving our knowledge on the details of CII in yeast and binding of inhibitors to the complex. I have only a few minor comments:

Page 6 and 16: It is stated that the Q site is partially occupied by endogenous ubiquinone in the native structure. Please clarify how it was established that there is partial occupancy. Is the density of poorer quality compared to the occupancy of the inhibitor in the bixafen structure? It would be beneficial to add the coulomb potential density map of UQ1 from the native structure in Supplementary Fig. 3 for comparison.

This point was also raised by another reviewer (R2) and indeed required greater clarity.

There is clear density in the CII-nat map to position the ubiquinone head at the Q site however, we don't see clear continuous density for its tail to allow modelling of the entire molecule. We have thus removed the word 'partially' and expanded this section slightly to clarify this point in the main manuscript as follows:

'In the CII-nat structure, the Q site is occupied by an endogenous ubiquinone molecule (UQ6, Supplementary Fig. 2 and 3b). Due to a lack of continuous density, the modelled UQ6 molecule is structurally equivalent to ubiquinone-1, with the remaining of the tail disordered most likely in the lipid phase; Supplementary Fig. 2 and 3b)

We have also added a panel for the cryo-EM density of the ubiquinone in Supplementary Fig. 3b and have added the map threshold to show density quality. Although at lower threshold, the Coulomb map is interpretable to position reliably the UQ molecule (now also changed from UQ1 to the endogenous UQ6). Its final position converged using real space refinement and is consistent with partial modelling of UQ in other species (ref 17,20,22).

Page 14, 16 and Supplementary Information: Reference to Supplementary Information Fig. 3 follows after reference to Supplementary Information Fig. 4 in the text. For better readability, please switch the numbers on the two figures.

This is a mistake which had escaped our attention.

We have been through the revised submission carefully and hope to now have in-text references to main figures and SI material in order of appearance.

Other modifications:

1. We changed the title to be within 15 words
2. We have shortened the abstract to fit within 150 words.